# Proportion of stroke types in Madagascar: A tertiary-level hospital-based case series

**Julia Riedmann**[1], **Andriamihaja Flavien Solonavalona**[2], **Adriamboahanginiaina Ravosoa Rakotozafy**[2], **Solofo Ralamboson**[2], **Matthias Endres**[1,3,4,5,6,7], **Bob Siegerink**[3,8], **Eberhard Siebert**[9], **Samuel Knauss**[1,7,10☉], **Julius Valentin Emmrich**[1,7,10☉]*

1 Department of Neurology, Charité—Universitätsmedizin Berlin, Berlin, Germany, 2 Soavinandriana Military Hospital (CENHOSOA), Antananarivo, Madagascar, 3 Center for Stroke Research, Charité—Universitätsmedizin Berlin, Berlin, Germany, 4 German Center for Neurodegenerative Diseases (DZNE), Partner Site Berlin, Göttingen, Germany, 5 German Centre for Cardiovascular Research (DZHK), Partner Site Berlin, Göttingen, Germany, 6 ExcellenceCluster NeuroCure, Berlin, Germany, 7 Berlin Institute of Health, Berlin, Germany, 8 Department of Clinical Epidemiology, Leiden University Medical Center, Leiden University, Leiden, The Netherlands, 9 Institute of Neuroradiology, Charité—Universitätsmedizin Berlin, Berlin, Germany, 10 Heidelberg Institute of Global Health, Heidelberg University, Heidelberg, Germany

☉ These authors contributed equally to this work.

* julius.emmrich@charite.de

**Data Availability Statement:** Two datasets including (1) medical record data and (2) CT imaging data are available under Creative Commons License v 4.0 (Attribution-ShareAlike) as

## Abstract

### Background

Like other countries in sub-Saharan Africa, Madagascar has a high burden of stroke. The Malagasy population is unique in sharing both African and Asian ancestry. The proportion of ischemic and hemorrhagic stroke types is unknown for this population.

### Aim

Our aim was to establish the proportion of stroke types and known risk factors for the Malagasy population.

### Methods

We conducted a single-center, tertiary-level hospital-based case series. We included all patients with a CT-imaging confirmed stroke who presented at the emergency ward of the study hospital between January 1, 2017, and November 20, 2018.

### Results

Of 223 patients with CT-confirmed stroke, 57.4% (128/223, 95% CI: 51–64%) had an ischemic stroke and 42.6% (95/223, 95% CI: 36–49%) had an intracranial hemorrhage. The majority (89.5%; 85/95, 95% CI: 83–96%) of intracranial hemorrhages were intracerebral; 4.2% (4/95, 95% CI: 0–8%) had a subdural hematoma, 5.3% (5/95, 95% CI: 1–10%) had a subarachnoid hemorrhage, there was one isolated intraventricular hemorrhage (1.1%; 1/95, 95% CI: -1-3%). The prevalence of hypertension among stroke patients was high (86.6%; 187/216, 95% CI: 82–91%).

open data on Figshare Repository in raw data format: (1) 10.6084/m9.figshare.14914737, (2) 10.6084/m9.figshare.14919654.

**Funding:** This study was financially supported by Theracur Foundation. JE and SK are participants in the BIH-Charité Digital Clinician Scientist Program funded by the Charité – Universitätsmedizin Berlin and the Berlin Institute of Health. The funders had no role in study design, data collection and analysis, decision to publish, or preparation of the manuscript.

**Competing interests:** I have read the journal´s policy and the authors of this manuscript have the following competing interests: ME received funding from DFG under Germany's Excellence Strategy – EXC-2049 – 390688087, BMBF, DZNE, DZHK, EU, Corona Foundation, and Fondation Leducq. ME reports grants from Bayer and fees paid to Charité from AstraZeneca, Bayer, Boehringer Ingelheim, BMS, Daiichi Sankyo, Amgen, GSK, Sanofi, Covidien, Novartis, Pfizer, all outside the submitted work. This does not alter our adherence to PLOS ONE policies on sharing data and materials.

## Conclusions

Our study is the first to report the proportion of stroke types and known risk factors in Madagascar. We find that the proportion of hemorrhagic strokes was unexpectedly higher than that reported from other countries in sub-Saharan Africa. Our findings highlight the need for a country-specific approach to stroke prevention, treatment, and rehabilitation and provide guidance on public health resource allocation in Madagascar.

## Introduction

Stroke, one of the leading causes of permanent disability and the second leading cause of death, accounts for 6 million deaths annually. Approximately 70% of these occur in low- and middle-income countries [1, 2]. Sub-Saharan Africa (SSA), home to around a fifth of the world's population, bears a high burden of stroke with an age-standardized incidence rate of up to 316 per 100,000, an age-standardized prevalence rate of 1,283 per 100,000, and a case fatality over one month of 24% [3–5]. In addition, stroke in SSA is increasingly affecting a younger age group and causes poorer long-term outcomes than in the developed world aggravating the social and economic toll of disease [6, 7]. As the population of SSA is the fastest growing and life expectancy is increasing most rapidly of all world regions, overall stroke prevalence is steadily rising [2, 8]. Strategies to reduce stroke burden and to adequately allocate health resources are urgently needed.

Despite the rising tide of stroke, stroke research productivity is low while epidemiology and proportion of stroke types are unknown in many countries in SSA [9]. The most recent estimate of stroke burden from the Global Burden of Disease (GBD) project included only 62 studies with participants from SSA, a mere 1.5% of the 4,058 studies used for analysis [2]. Likewise, INTERSTROKE, the largest case-control study on stroke risk factors to date, included only 3.6% of participants from SSA indicating challenges to follow-up and lack of health facilities in which computed tomography (CT) scan or magnetic resonance imaging (MRI) were available [9]. The World Health Organization, United States' National Academy of Sciences as well as the recently inaugurated African Stroke Organization urgently call for improving local stroke data in SSA [10–12].

In Madagascar, one of the world's least developed countries with a population of 27 million [13], the estimated life-time risk of stroke is 15.5% [14]. According to the Ministry of Health, stroke is the most common reason for in-hospital death albeit less than 5% of causes of deaths in Madagascar are registered [15, 16]. To date, the proportion of stroke types has not been described for the Malagasy population.

Our aim was to characterize the proportion of stroke types in a hospital-based case series of imaging-confirmed stroke patients. In addition, we describe the prevalence of known risk factors and fatality rates.

## Methods

### Study setting

This study was conducted at Soavinandriana Military Hospital in Antananarivo, a 454-bed national referral hospital. A CT scanner was available 24/7. Thrombolytic therapy was not available. There was no dedicated stroke unit. Healthcare at the hospital was free for civil servants and their families.

## Study design

This was a retrospective hospital-based study including all patients with a CT-imaging confirmed stroke and accessible patient files who presented at the emergency ward of the hospital between January 1, 2017, and November 20, 2018.

## Case finding methods and inclusion criteria

We identified cases based on the hospital's emergency room register, which contained a brief medical history of all patients who sought admission for an acute illness. We extracted all cases that had a neurological deficit of sudden onset including weakness, sensory loss or inattention, speech disturbances (dysarthria or aphasia), visual problems, limb ataxia and gait unsteadiness, as well as non-specific signs including dizziness, seizures, loss of consciousness, impaired cognitive function, and thunderclap headache. We retrieved the medical records of patients with at least one of these symptoms and who were subsequently admitted to the hospital. Of those, we included all patients whose medical record contained CT images of the brain showing an ischemic or hemorrhagic stroke.

## Exclusion criteria

Patients whose medical records or CT images could not be retrieved from hospital archives were excluded.

## Data collection and data entry

**Medical records.** We digitized medical records using a digital camera (EOS 550D DSLR, Canon). Data were entered into a standardized data collection form by three trained data collectors (JR, AS, RA). Data extracted from medical records included sociodemographic characteristics (sex and age), clinical characteristics (symptoms upon arrival, time of symptom onset, admission to the ER, and CT scan, duration of hospital stay, and in-hospital mortality), ultrasound findings (echocardiography and carotid duplex scan), lab tests (glycated hemoglobin, lipid profile), medical history, and risk factors (hypertension, diabetes, body mass index, tobacco- and alcohol-consumption, family history of stroke). Data quality was continuously monitored by a supervisor who trained the data collection team. Data were crosschecked and screened for double entries, out of range values, and overall consistency. We anonymized data at the data entry level to protect participants' personal identifiable information.

**Imaging data acquisition and interpretation.** Nonenhanced CT was performed on a multidetector CT scanner (SOMATOM Perspective CT VC40, Siemens). Images were developed on X-ray film. We visualized and digitized those images using an X-ray film viewer and a high-resolution digital camera (EOS 550D DSLR, Canon). Digitized images were read by a neuroradiologist (ES) who entered the scan results into a standardized data collection form. Data extracted from CT images included stroke subtype (ischemic or hemorrhagic), lesion side (right, left or bilateral), age (acute, < 24 hours; subacute, 1–5 days; or chronic (> 5 days), lesion size (ischemic: lacunar, < 2/3 of vascular territory, > 2/3 of vascular territory; hemorrhagic: intracerebral hemorrhage volume < 30 ml, intracerebral hemorrhage > 30 ml), lesion expansion, previous lesions and white matter lesions (categorized according to the Fazekas scale (0, no lesions; 1, punctuate lesions; 2, beginning confluence of lesions; 3, large confluent areas)) [17].

Ischemic strokes were classified by vascular territory into anterior cerebral, middle cerebral, posterior cerebral, and vertebrobasilar artery strokes as well as strokes affecting more than one vascular territory and by etiology (i.e., cardioembolism, small-vessel disease or undetermined etiology). Intracranial hemorrhages were classified by location into typical (affecting the basal

ganglia, thalamus, pons, or cerebellum) and atypical (all other locations) intracerebral hemorrhage as well as subarachnoid hemorrhage, and subdural hematoma.

**Definitions of risk factors.** Arterial hypertension, tobacco- and alcohol-consumption, and a family history of stroke were self-reported risk factors. Overweight was recorded as a risk factor if body-mass-index (BMI) was more than 25. Diabetes was either self-reported or was newly diagnosed during hospitalization (glycated hemoglobin > 6,5%). Hyperlipidemia was defined as blood lipid levels above the upper reference threshold of the hospital's laboratory.

**Data analysis.** We used descriptive statistics to summarize the data set and independent t-tests to compare metric variables and Pearson's Chi-Square for categorical variables. Analyses were performed in SPSS (IBM SPSS Statistics, Version 25, 2017).

**Ethics approval and consent to participate.** This study was approved by the Institutional Review Board of Soavinandriana Hospital (067/CENHOSOA/DG/DT) on June 28, 2018. Informed consent was waived.

## Results

We included a total of 223 patients with CT confirmed stroke diagnosis. Lipid profiles were available for 62.3% (139/223, 95% CI: 56–69%) of patients, 29.1% (65/223, 95% CI: 23–35%) had HbA1c measurements, whereas cardiac echo was performed in 24.2% (54/223, 95% CI: 19–30%) and carotid duplex sonography in 9.9% (22/223, 95% CI: 6–14%) of patients.

Medical history was assessed for hypertension in 96.9% (216/223, 95% CI: 95–99%), diabetes in 80.7% (180/223, 95% CI: 76–86%), and tobacco consumption in 68.6% (153/223, 95% CI: 63–75%) of patients.

### Clinical and demographic characteristics

Demographic and clinical characteristics are summarized in Table 1 stratified according to stroke type. Patients with hemorrhagic strokes were younger (60 (51–67) vs. 64 (58–72) years, p = 0.003) and more likely to be male (69.5% (95% CI: 60–79%) vs. 55.5% (95% CI: 47–64%), p = 0.034) than patients with ischemic strokes. Most patients (71.8%; 160/223, CI: 66–78%) arrived later than 6 hours after symptom onset; a first CT scan was obtained 3.2 (2.2–5.7) hours after initial presentation. Fig 1 summarizes clinical characteristics upon presentation by stroke type.

### Imaging characteristics

Of 223 patients, 57.4% (128/223, 95% CI: 51–64%) had an ischemic and 42.6% (95/223, 95% CI: 36–49%) a hemorrhagic stroke (Fig 2). Of ischemic and hemorrhagic strokes combined, 45.7% (102/223, 95% CI: 39–52%) were in the left and 41.7% (93/223, 95% CI: 35–48%) in the right hemisphere. One in eight (12.5%, 28/223, 95% CI: 8–17%) was located bilaterally.

### Ischemic stroke

Vascular territories affected by ischemic strokes (128/223) are depicted in Fig 2. We found strokes affecting more than two-thirds of a vascular territory in 29.7% (38/128, 95% CI: 22–38%), less than two-thirds in 50% (64/128, 95% CI: 41–59%), and lacunar strokes in 20.3% (26/128, 95% CI: 13–27%) of patients. Based on imaging, stroke etiology was undetermined in most patients (75.5%; 96/128, 95% CI: 67–83%). Imaging was suggestive of small vessel occlusion in 15.6% (20/128, 95% CI: 9–22%) and cardioembolism as the cause of stroke in 9.4% (12/128; 95% CI: 4–15%) of patients. Almost half of patients 39.4% (50/127, 95% CI: 31–48%) had lesions consistent with previous strokes. White matter lesions indicating chronic small vessel

**Table 1. Demographic and clinical characteristics by stroke type.**

| | Total[a] | IS[a] | HS[a] | p-value[b] |
|---|---|---|---|---|
| **Total** | **223 (100)** | **128 (57.4)** | **95 (42.6)** | |
| Sex | 223 | 128 | 95 | |
| female | 86 (38.6) | 57 (44.5) | 29 (30.5) | 0.034 |
| male | 137 (61.4) | 71 (55.5) | 66 (69.5) | |
| Age | 223 | 128 | 95 | |
| median age; years (IQR) | 62 (54–69) | 64 (58–72) | 59 (51–67) | 0.003 |
| 24–35 | 2 (0.9) | 2 (1.6) | 0 | |
| 36–50 | 32 (14.3) | 9 (7.0) | 23 (24.2) | |
| 51–65 | 99 (44.4) | 57 (44.5) | 42 (44.2) | |
| 66–80 | 70 (31.4) | 47 (36.7) | 23 (24.2) | |
| 81–94 | 20 (9.0) | 13 (10.2) | 7 (7.4) | |
| Time between symptom onset and arrival at ER | 223 | 128 | 95 | |
| <3h | 36 (16.1) | 16 (12.5) | 20 (21.1) | |
| 3–4.5h | 15 (6.7) | 9 (7.0) | 6 (6.3) | |
| 4.5-6h | 12 (5.4) | 4 (3.1) | 8 (8.4) | |
| 6-24h | 86 (38.6) | 52 (40.6) | 34 (35.8) | |
| 24-72h | 36 (16.1) | 23 (18.0) | 13 (13.7) | |
| 72h-7d | 22 (9.9) | 12 (9.4) | 10 (10.5) | |
| >7d | 16 (7.2) | 12 (9.4) | 4 (4.2) | |
| Time between ER admission and CT scan | 223 | 128 | 95 | |
| median time; hours (IQR) | 3.2 (2.2–5.7) | 3.2 (2.2–6.3) | 3.2 (2.3–5.4) | 0.826 |
| Length of hospital stay | 164 | 104 | 60 | |
| median length; days (IQR) | 12.5 (8.2–18.2) | 11.8 (9.4–18.4) | 15.6 (13.3–21.4) | 0.868 |
| < 7d | 16 (9.8) | 12 (11.5) | 4 (6.7) | |
| 7d–2w | 69 (42.1) | 54 (51.9) | 15 (25.0) | |
| 2–4w | 65 (39.6) | 26 (25.0) | 39 (65.0) | |
| > 4w | 14 (8.5) | 12 (11.5) | 2 (3.3) | |
| Death | 223 | 128 | 95 | |
| yes | 49 (22.0) | 22 (17.2) | 27 (28.4) | 0.045 |
| Length of hospitalization until death | 48 | 22 | 26 | |
| median length; days (IQR) | 6.9 (3.2–12.9) | 12.8 (5.6–15.3) | 4.6 (1.4–10.0) | 0.179 |
| < 24h | 4 (8.3) | 0 | 4 (15.4) | |
| 24h–7d | 21 (43.8) | 9 (40.9) | 12 (46.2) | |
| 7d–2w | 13 (27.1) | 6 (27.3) | 7 (26.9) | |
| > 2w | 10 (20.8) | 7 (31.8) | 3 (11.5) | |

IS = ischemic stroke; HS = hemorrhagic stroke

IQR = interquartile range

[a] Number and (%), if not indicated otherwise

[b] Statistical tests: chi-square test of independence

disease were present in 71.1% (91/128, 95% CI: 63–79%) of patients. Table 2 summarizes the imaging characteristics of patients with ischemic stroke.

## Intracranial hemorrhage

Of 95 intracranial hemorrhages on CT imaging, 89.5% (85/95, 95% CI: 83–96%) were classified as intracerebral hemorrhage, 5.3% (5/95, 95% CI: 1–10%) as subarachnoid hemorrhage, 4.2%

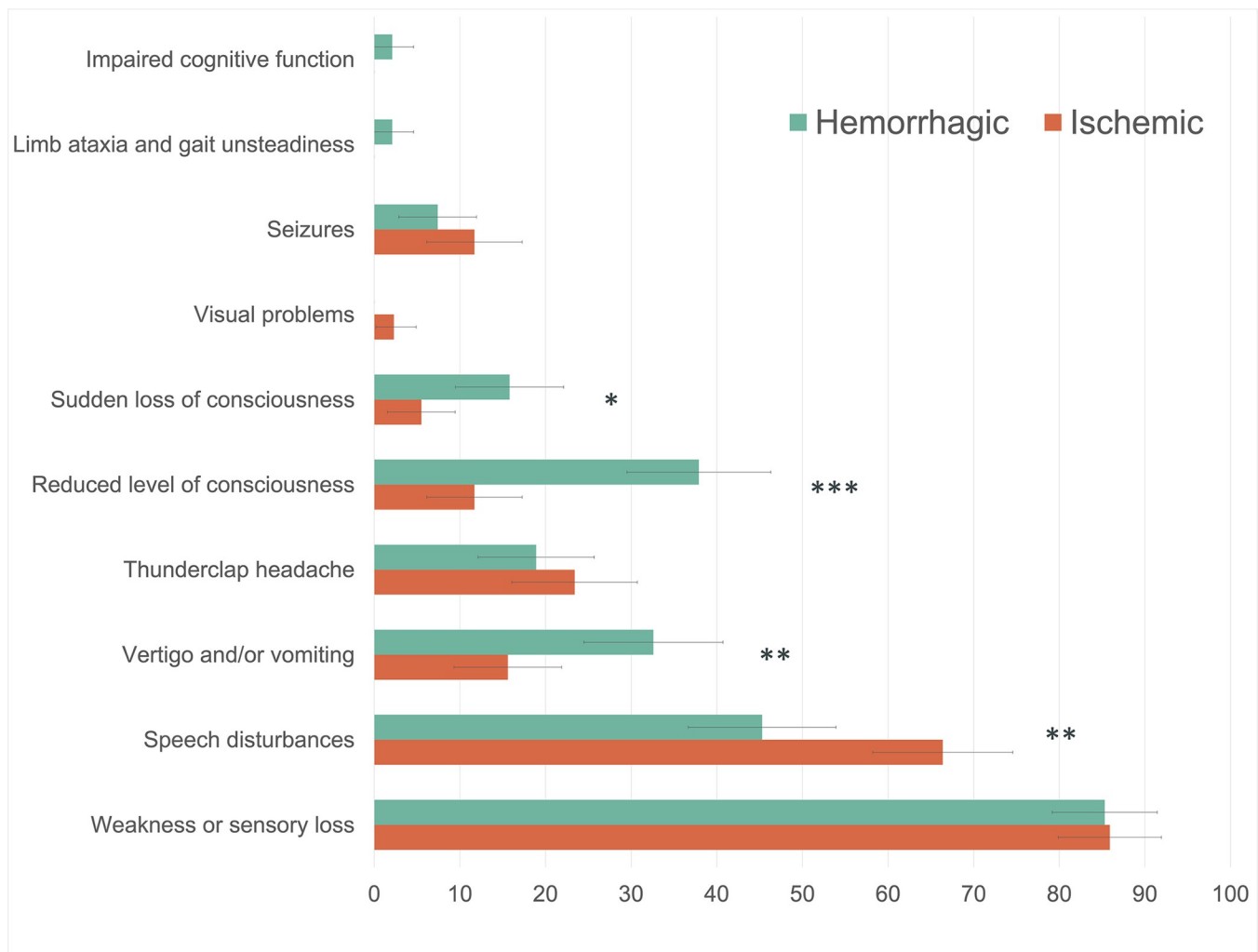

**Fig 1. Symptoms of patients with CT-confirmed stroke upon presentation in the emergency room by stroke type.** Chi-square: * p<0.05; ** p<0.01; *** p<0.001.

(4/95, 95% CI: 0–8%) as subdural hemorrhage, and one patient (1.1%; 1/95, 95% CI: -1-3%) had an isolated intraventricular hemorrhage without intraparenchymal hemorrhage. Most intracerebral hemorrhages (76,5%; 65/85, 95% CI: 67–86%) were in the basal ganglia, thalamus, pons, or cerebellum, which are typical for a hypertensive etiology. Large hemorrhages with volumes >30ml occurred in 28.0% (26/90, 95% CI: 19–38%) and intraventricular hemorrhages in 41.5% (39/94, 95% CI: 31–52%) of patients. Table 3 summarizes the imaging characteristics of patients with hemorrhagic stroke.

## Ultrasound imaging

S1 Table in S1 File summarizes echocardiographic and carotid duplex sonography findings by stroke type.

## In-hospital mortality

Forty-nine patients died during the hospital stay (22.0%; 49/223, 95% CI: 17–28%). In-hospital mortality differed significantly between ischemic stroke (17.2%; 22/128, 95% CI: 11–24%) and

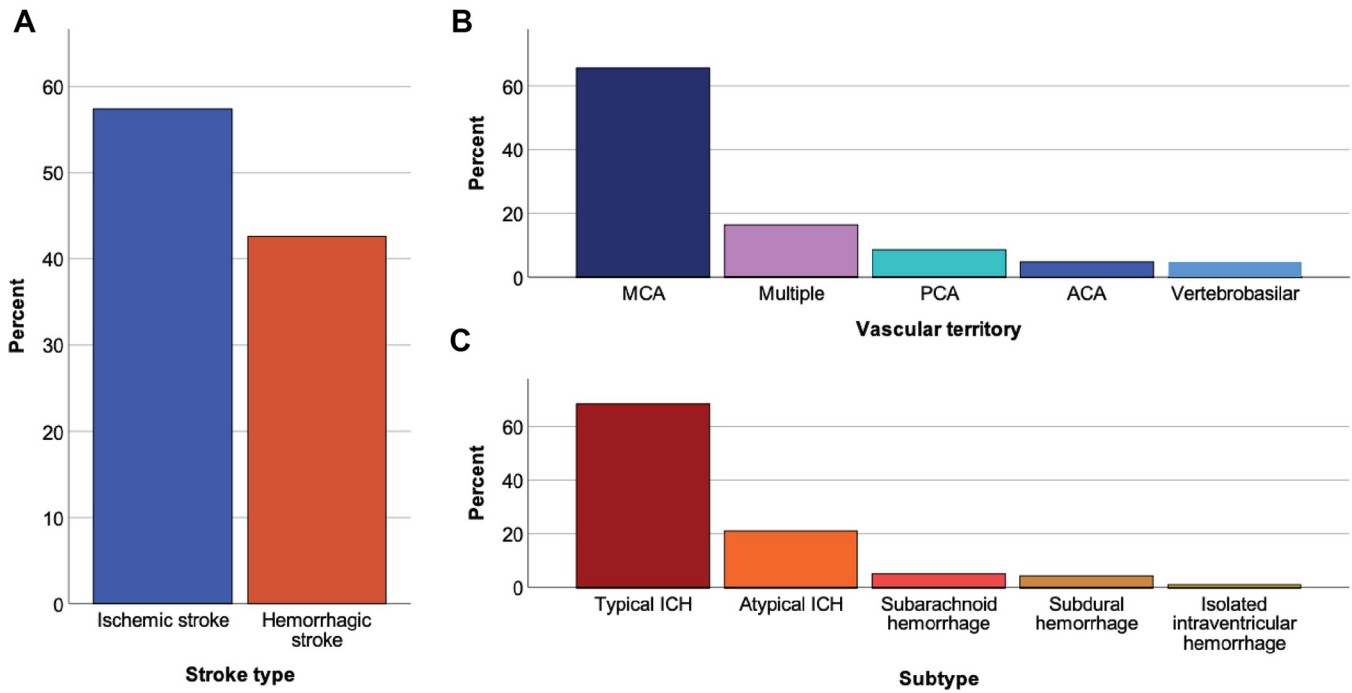

**Fig 2. Radiological classification of ischemic and hemorrhagic stroke types.** Panel A depicts the distribution to ischemic vs. hemorrhagic stroke as percentage of all strokes, panel B and C show distribution of vascular territories and stroke subtypes separately for ischemic (**B**) and hemorrhagic stroke (**C**). ACA, anterior cerebral artery; MCA, middle cerebral artery; PCA, posterior cerebral artery; ICH, intracerebral hemorrhage.

hemorrhagic stroke (28.4%; 27/95, 95% CI: 19–38%, p = 0.045, Table 1). Most in-hospital deaths were directly attributed to the brain lesion (63.3%; 31/49, 95% CI: 49–77%). Secondary causes of death included chest infection, respiratory failure and cardiac infarction.

## Risk factors

Hypertension was the most prevalent risk factor across all groups 187/216 (86.6%; 187/216, 95% CI: 82–91%), followed by active alcohol consumption 38.8% (59 /152, 95% CI: 31–47%), tobacco consumption 29.4% (45/153, 95% CI: 22–37%), and diabetes 17.8% (32/180, 95% CI: 12–23%). Alcohol consumption was more common among patients with a hemorrhagic stroke (64.0% (48/75, 95% CI: 53–75%) vs. 45.5% (35/77, 95% CI: 34–57%), p = 0.022); diabetes was more common among patients with ischemic stroke (23.6% (25 /106, 95% CI: 15–32%) vs. 9.5% (7/74, 95% CI: 3–16%), p = 0.015). Blood lipid profiles revealed dyslipidemia in 30.2% (42/139, 95% CI: 23–38%) of patients. One-third 34.7% (25/72, 95% CI: 24–46%) were overweight (body-mass-index >25). Table 4 summarizes contributing risk factors among patients with ischemic or hemorrhagic stroke.

## Discussion

Our study describes the proportion of stroke types in Madagascar using a hospital-based, imaging-confirmed series of cases. The conspicuous strengths of our study were the inclusion of stroke patients irrespective of an individual's financial situation to reduce the risk of selection bias and use of a standardized image analysis protocol.

Among our study population, the majority had ischemic strokes (128/223; 57.4%) predominantly in the middle cerebral artery territory followed by lacunar strokes. Hemorrhagic strokes

**Table 2. Imaging characteristics of patients with ischemic stroke.**

| | Total[a] |
|---|---|
| | **128 (100)** |
| Side of stroke lesion | 128 |
| left | 56 (43.8) |
| right | 56 (43.8) |
| both | 16 (12.5) |
| Age of lesion | 128 |
| acute (< 24h) | 73 (57.0) |
| subacute (24h-5d) | 14 (10.9) |
| chronic (> 5d) | 39 (30.5) |
| unclear | 2 (1.6) |
| Vascular territory | 128 |
| ACA | 6 (4.7) |
| MCA | 84 (65.6) |
| PCA | 11 (8.6) |
| vertebro-basilar | 6 (4.7) |
| multiple | 21 (16.4) |
| Size of lesion | 128 |
| lacunar | 26 (20.3) |
| < 2/3 of territory | 64 (50.0) |
| > 2/3 of territory | 38 (29.7) |
| Stroke subtype | 128 |
| cardioembolism (embolic stroke) | 12 (9.4) |
| small-vessel occlusion (lacune) | 20 (15.6) |
| undetermined etiology | 96 (75.0) |
| Previous stroke on CT | 50/127 (39.4) |
| White matter lesions | 128 |
| Fazekas 0 | 37 (28.9) |
| Fazekas 1 | 35 (27.3) |
| Fazekas 2 | 26 (20.3) |
| Fazekas 3 | 30 (23.4) |

ACA = anterior cerebral artery; MCA = middle cerebral artery; PCA = posterior cerebral artery

[a] Number and % if not indicated otherwise

accounted for 95/223 (42.6%) of cases and the majority occurred in locations typical for hypertensive intracerebral hemorrhage. Almost 90% of stroke patients had hypertension, underscoring the importance of implementing effective prevention strategies to reduce stroke burden. Around a third of ischemic and hemorrhagic strokes were severe, affecting more than 2/3 of a vascular territory or exceeding a bleeding volume of 30ml indicating high levels of functional disability and emphasizing the need for stroke rehabilitation. The median age of our study population was 62 years and patients were more likely to be male (61.4%), exemplifying the substantial economic impact of stroke in Madagascar by affecting a relatively young and productive population.

Scientific literature on stroke in Madagascar is scarce. The Pubmed/MEDLINE database contains only six publications using the search terms "stroke" and "Madagascar", three of which are hospital-based case series. None included an unselected sample of imaging-confirmed stroke patients. Razafindrasata et al. included only patients with acute motor deficits;

**Table 3. Imaging characteristics of patients with hemorrhagic stroke.**

| | Total[a] |
|---|---|
| | **95 (100)** |
| Side of stroke lesion | 95 |
| left | 46 (48.4) |
| right | 37 (38.9) |
| both | 12 (12.6) |
| Age of lesion | 95 |
| acute (< 24h) | 90 (94.7) |
| subacute (24h-5d) | 5 (5.3) |
| chronic (> 5d) | 0 |
| unclear | 0 |
| Intracerebral hemorrhage | 85/95 (89.5) |
| Location of origin | 85 |
| typical | 65 (76.5) |
| atypical | 20 (23.5) |
| Intraventricular hemorrhage | 39/94 (41.5) |
| Infratentorial origin of hemorrhage | 8/95 (8.4) |
| ICH volume >30ml | 26/90 (28.9) |
| Subarachnoid hemorrhage | 5/95 (5.3) |
| Subdural hemorrhage | 4/95 (4.2) |

ICH = intracerebral hemorrhage

[a] Number and (%), if not indicated otherwise

150 of 227 patients had CT imaging, 45% of those were hemorrhagic strokes [18]. Rasaholiarison et al. included only patients with lacunar strokes; all 83 patients had CT imaging, 67% of those were hemorrhagic strokes [19]. Stenumgård et al. report clinical characteristics, socio-demographic factors, and outcomes in 30 consecutive stroke patients but only 3 of those had a CT [20]. Remaining publications on stroke in Madagascar are case reports [21–23].

The Stroke Investigative Research and Educational Network (SIREN) study, the largest study on the proportion of stroke types and associated risk factors in SSA to date, included 2,118 consecutive case-control pairs from Ghana and Nigeria [24]. INTERSTROKE, an international case-control study included 973 stroke patients from Mozambique, Nigeria, South Africa, Sudan, and Uganda [25]. Compared to results from SIREN and INTERSTROKE, stroke patients in Madagascar were older (59.0 and 58.7 vs. 62.1 years) and had a higher likelihood of hemorrhagic stroke (32% and 30.2% vs. 42.6%); hypertension was the most common risk factor in all studies. Hypertension, the most important risk factor for stroke, is common in Madagascar but not more common than in other countries in SSA. Using previous guideline recommendations, 27.0% and 29.7% of rural and urban populations in Madagascar have hypertension defined as blood pressure readings greater than 140/90 mm Hg [26]. This is similar to the prevalence of hypertension found in rural and peri-urban populations in Uganda, South Africa, Tanzania, and Nigeria [27]. Taken together, the higher rate of hemorrhagic strokes in our study might be caused by other modifiable risk factors or genetic predisposition to intracerebral hemorrhage [28, 29].

Compared to other hospital-based case series in SSA ranging from 25.9 to 41.1% of all stroke patients [30–32], the fatality rate in our study (22.0%) was low. This indicates either successful treatment or selection bias which might have been caused excluding patients who died in the emergency room before being admitted to hospital.

**Table 4. Contributing risk factors among patients with ischemic or hemorrhagic stroke.**

| | Total[a] | IS[a] | HS[a] | p-value[b] |
|---|---|---|---|---|
| **Total** | **223 (100)** | **128 (57.4)** | **95 (42.6)** | |
| Stroke family history | 42 | 24 | 18 | |
| | 11/42 (26.2) | 3/24 (12.5) | 8/18 (44.4) | 0.020 |
| Tobacco | 153 | 85 | 68 | |
| non-smoker | 71 (46.4) | 38 (44.7) | 33 (48.5) | 0.411 |
| active smoker | 45 (29.4) | 23 (27.1) | 22 (32.4) | |
| former smoker | 37 (24.2) | 24 (28.2) | 13 (19.1) | |
| Alcohol | 152 | 77 | 75 | |
| no consumption | 69 (45.4) | 42 (54.5) | 27 (36.0) | 0.012 |
| active consumption | 59 (38.8) | 21 (27.3) | 38 (50.7) | |
| former consumption | 24 (15.8) | 14 (18.2) | 10 (13.3) | |
| Hypertension | 216 | 127 | 89 | |
| | 187 (86.6) | 113 (89.0) | 74 (83.1) | 0.216 |
| Diabetes | 180 | 106 | 74 | |
| | 32 (17.8) | 25 (23.6) | 7 (9.5) | 0.015 |
| BMI | 72 | 43 | 29 | |
| median; BMI (IQR) | 23.5 (20.3–25.8) | 24.2 (19.5–25.7) | 22.0 (20.6–26.5) | 0.570 |
| <18.5 | 9 (12.5) | 8 (18.6) | 1 (3.4) | |
| 18.5–25.0 | 38 (52.8) | 19 (44.2) | 19 (65.5) | |
| 25.0–30.0 | 18 (25.0) | 10 (23.3) | 8 (27.6) | |
| 30.0–35.0 | 6 (8.3) | 5 (11.6) | 1 (3.4) | |
| >35.0 | 1 (1.4) | 1 (2.3) | 0 | |
| HbA1c | 65 | 45 | 20 | |
| normal | 32 (49.2) | 19 (42.2) | 13 (65.0) | 0.090 |
| increased | 33 (50.8) | 26 (57.8) | 7 (35.0) | |
| Total cholesterol | 157 | 96 | 61 | |
| normal | 134 (85.4) | 81 (84.4) | 53 (86.9) | 0.681 |
| decreased | 9 (5.7) | 5 (5.2) | 4 (6.6) | |
| increased | 14 (8.9) | 10 (10.4) | 4 (6.6) | |
| LDL cholesterol | 139 | 83 | 56 | |
| normal | 97 (69.8) | 55 (66.3) | 42 (75.0) | 0.185 |
| increased | 42 (30.2) | 28 (33.7) | 14 (25.0) | |
| HDL cholesterol | 142 | 84 | 58 | |
| normal | 63 (44.4) | 35 (41.7) | 28 (48.3) | 0.834 |
| decreased | 79 (55.6) | 49 (58.3) | 30 (51.7) | |
| Triglycerides | 156 | 96 | 60 | |
| normal | 133 (85.3) | 83 (86.5) | 50 (83.3) | 0.585 |
| decreased | 3 (1.9) | 1 (1.0) | 2 (3.3) | |
| increased | 20 (12.8) | 12 (12.5) | 8 (13.3) | |

IS = ischemic stroke; HS = hemorrhagic stroke

BMI = body mass index (kg/m$^2$); HbA1c = glycated hemoglobin; LDL = low-density lipoprotein; HDL = high-density lipoprotein

IQR = interquartile range

[a] Number and % if not indicated otherwise

[b] Statistical tests: chi-square test of independence

Our study has limitations. First, the results of our single-center study in an urban setting might not reflect the true community burden of stroke. However, access to healthcare including CT scans was free of charge at the study hospital for patients reducing selection bias otherwise introduced by a households' ability to pay [33]. In addition, prompt imaging and comprehensive investigation would not have been feasible in a community-setting. Second, the study hospital being a tertiary-level referral hospital might have introduced a selection bias towards more severe cases of stroke, which might explain the relatively high mortality rate. Third, by including CT-confirmed strokes only, minor strokes for which a CT scan might not have been performed or posterior circulation strokes, for which CT imaging is known to be less sensitive, might be underrepresented. Fourth, while medical records including demographic and clinical data were available for all patients, they were not equally thorough and self-reported medical history may lead to an underestimation of negative findings. On the other hand, patients' medical history was consistently assessed for relevant risk factors, like hypertension, in 96.9% of records. Fifth, clinical data on stroke severity was limited and no standardized data on stroke severity was available. Nevertheless, we used imaging characteristics to determine stroke severity. Last, our study was not designed to assess stroke prevalence or incidence, as the details of the source population from which the study hospital draws its patients from were unknown.

In conclusion, our study results contribute to determining the clinical outcome and prognosis of stroke patients in Madagascar and provide guidance on public health resource allocation for stroke prevention, treatment, and rehabilitation. In addition, our results may encourage community-based stroke surveillance studies to be conducted in Madagascar.

## Supporting information

**S1 File. Ultrasound imaging by stroke type.**
(PDF)

## Author Contributions

**Conceptualization:** Julia Riedmann, Solofo Ralamboson, Matthias Endres, Bob Siegerink, Samuel Knauss, Julius Valentin Emmrich.

**Data curation:** Julia Riedmann, Andriamihaja Flavien Solonavalona, Adriamboahanginiaina Ravosoa Rakotozafy, Eberhard Siebert.

**Formal analysis:** Julia Riedmann, Eberhard Siebert, Julius Valentin Emmrich.

**Funding acquisition:** Matthias Endres, Samuel Knauss, Julius Valentin Emmrich.

**Investigation:** Samuel Knauss, Julius Valentin Emmrich.

**Methodology:** Julia Riedmann, Bob Siegerink, Samuel Knauss, Julius Valentin Emmrich.

**Project administration:** Samuel Knauss, Julius Valentin Emmrich.

**Resources:** Solofo Ralamboson.

**Supervision:** Samuel Knauss, Julius Valentin Emmrich.

**Writing – original draft:** Julia Riedmann, Samuel Knauss, Julius Valentin Emmrich.

**Writing – review & editing:** Samuel Knauss, Julius Valentin Emmrich.

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
