## [Decision Letter · Decision Letter 0]

7 Jul 2022

PONE-D-22-13670Proportion of stroke types in Madagascar: a tertiary-level hospital-based case seriesPLOS ONE

Dear Dr. Emmrich,

Thank you for submitting your manuscript to PLOS ONE. After careful consideration, we feel that it has merit but does not fully meet PLOS ONE’s publication criteria as it currently stands. Therefore, we invite you to submit a revised version of the manuscript that addresses the points raised during the review process.

We look forward to receiving your revised manuscript.

Kind regards,

Ismail Ibrahim Ismail, MD

Academic Editor

PLOS ONE

Journal Requirements:

"I have read the journal´s policy and the authors of this manuscript have the following competing interests: ME received funding from DFG under Germany's Excellence Strategy – EXC-2049 – 390688087, BMBF, DZNE, DZHK, EU, Corona Foundation, and Fondation Leducq. ME reports grants from Bayer and fees paid to Charité from AstraZeneca, Bayer, Boehringer Ingelheim, BMS, Daiichi Sankyo, Amgen, GSK, Sanofi, Covidien, Novartis, Pfizer, all outside the submitted work."

Reviewers' comments:

Reviewer's Responses to Questions

**Comments to the Author**

1. Is the manuscript technically sound, and do the data support the conclusions?

Reviewer #1: Yes

Reviewer #2: Yes

Reviewer #3: No

2. Has the statistical analysis been performed appropriately and rigorously? 

Reviewer #1: Yes

Reviewer #2: Yes

Reviewer #3: No

3. Have the authors made all data underlying the findings in their manuscript fully available?

Reviewer #1: Yes

Reviewer #2: Yes

Reviewer #3: No

4. Is the manuscript presented in an intelligible fashion and written in standard English?

Reviewer #1: Yes

Reviewer #2: Yes

Reviewer #3: No

5. Review Comments to the Author

Reviewer #1: I do not think that this study represents the true stroke figures. The authors included patients with CT-confirmed stroke. A significant proportion of stroke patients would have a small infarction in MRI, which would not appear in CT. I think the authors should have included stroke patients meeting the old WHO definition of stroke, irrespective of the neuroimaging: acute-onset focal neurological deficits lasting more than 24hours. This would increase the proportion of ischemic stroke patients. Please consider adding this data to your study. Moreover, I think a significant proportion of stroke patients in SSA would not present to the hospital, which would explain the relatively higher prevalence of hemorrhagic stroke in this study.

"Sub-Saharan Africa (SSA), home to around a fifth of the world’s population, bears a high burden of stroke with an age-standardized incidence rate of 160 per 100,000" Actually, this is not a high incidence. The incidence of stroke in Germany is around 250/100,000 per year

"As the population of SSA is the 73 fastest growing and fastest ageing of all world regions" Please clarify this sentence.

In table S1: Please explain what do you mean by the following terminologies: instable plaque, thrombus, reduced blood flow velocity

Minor mistake: table 1 is described as figure 1

Reviewer #2: Very interesting well- written article that discussed the proportion of stroke types among Malagasy population

I have some comments that might help improve this nice article.

Line 164: (BMI) was more than 25 kg/m2, being an index it should not have units

Table 1: Length of hospital stay (days) (164, 104, 60) is a bit confusing → (days) should be (no of patients ) Then (days) should be in the next rows, same applied for Length of hospitalization until death

Table 2: side of lesion was bilateral in 12.5% of cases however 70% of cases has Fazekas I -III which means that 70 % of patients have bilateral lesions. So, what do you mean here by bilateral lesions?

You did not mention the causes of death in your patients (chest infection , respiratory failure , pulmonary embolism, expanding hge/ inf etc). This is very important to know because your mortality rate is relatively high 22%.

What about the other important stroke risk factors like ischemic heart disease, oral contraceptive pills, migraine, atrial fibrillation, anticoagulants ?

You should mention in the limitation section that being a tertiary-level hospital this may cause selection bias as you will receive more of complicated and severe cases and this might explain the relatively higher morality rate among your cohort.

Reviewer #3: This single center retrospective study using CT-confirmed strokes only, could miss a lot of ischemic strokes like minor strokes , posterior circulation strokes and others for which a CT scan might not have been less useful

Even demographic and clinical data were not equally thorough and self-reported medical history may lead to an underestimation of negative findings.

clinical data on stroke severity was limited and no standardized data on stroke severity was available.

All above mentioned might shed great doubt on the message of the study

6. PLOS authors have the option to publish the peer review history of their article (what does this mean?). If published, this will include your full peer review and any attached files.

Reviewer #1: **Yes: **Ahmed Elhfnawy

Reviewer #2: No

Reviewer #3: **Yes: **Ossama yassin mansour

---

## [Author Response · Author response to Decision Letter 0]

14 Aug 2022

Dear Dr Chenette, 

We would like to thank you and the reviewers for the constructive and insightful comments on our previous submission. By incorporating the editor’s and reviewers’ suggestions, we believe that we were able to significantly improve the quality of this manuscript. Please find on the following pages a point-by-point response to each of the comments. 

Reviewer #1: 

1. I do not think that this study represents the true stroke figures. The authors included patients with CT-confirmed stroke. A significant proportion of stroke patients would have a small infarction in MRI, which would not appear in CT. I think the authors should have included stroke patients meeting the old WHO definition of stroke, irrespective of the neuroimaging: acute-onset focal neurological deficits lasting more than 24hours. This would increase the proportion of ischemic stroke patients. Please consider adding this data to your study. 

Response: We thank the reviewer for pointing this out and fully agree with this comment. As non-contrast head CT is known to have low sensitivity for detecting ischemic strokes overall and performs even worse in patients with posterior fossa stroke, it would have been preferable to include patients based on the old WHO definition of stroke. However, the overall accuracy and completeness of the available medical records was poor. Most records did not specifically mention the suspicion or diagnosis of a stroke and did not contain standardized information on stroke severity or the clinical course during admission. Thus, using the old WHO definition of stroke would have drastically reduced our sample size and the overall reliability of results. Therefore, we identified cases based on the hospital’s emergency room register, from which we extracted all cases that had a neurological deficit of sudden onset. Of those, we included all patients whose medical record contained CT images of the brain showing an ischemic or hemorrhagic stroke. A particular strength of our study was that CT findings were not obtained from reports, but all images were digitized and read by an experienced neuroradiologist who entered the scan results into a standardized data collection form prior to analysis.

To address the reviewer’s concern the following sentence is included in the limitation section of the manuscript (page 20, line 327-329): 

“Third, by including CT-confirmed strokes only, minor strokes for which a CT scan might not have been performed or posterior circulation strokes, for which CT imaging is known to be less sensitive, might be underrepresented.”

2. Moreover, I think a significant proportion of stroke patients in SSA would not present to the hospital, which would explain the relatively higher prevalence of hemorrhagic stroke in this study.

Response: We fully agree with the comment that in general, only a minority of patients after stroke can be expected to present to a hospital in SSA. This might in particular affect more severe stroke cases. However, financial constraints are one of the main reasons for patients to forgo or delay care in SSA [1, 2]. Thus, a conspicuous strength of our study was the inclusion of stroke patients irrespective of an individual’s financial situation which is unique in Madagascar and likely reduced this selection bias. 

3. "Sub-Saharan Africa (SSA), home to around a fifth of the world’s population, bears a high burden of stroke with an age-standardized incidence rate of 160 per 100,000" Actually, this is not a high incidence. The incidence of stroke in Germany is around 250/100,000 per year.

Response: We thank the reviewer for spotting this error and apologize for our oversight. The age-standardized incidence rate is up to 316 per 100,000 for Sub-Saharan Africa. We have amended the sentence as follows (page 4, line 67-69): 

"Sub-Saharan Africa (SSA), home to around a fifth of the world’s population, bears a high burden of stroke with an age-standardized incidence rate of up to 316 per 100,000 [3]."

4. "As the population of SSA is the fastest growing and fastest ageing of all world regions..." Please clarify this sentence.

Response: We apologize for the lack of clarity. Sub-Saharan Africa (SSA) has a very high fertility rate of 4.7 births per woman compared to 2.4 births worldwide [4]). At the same time, life expectancy is increasing more rapidly in SSA than anywhere else [5]. To improve clarity, we have amended the sentence as follows (page 4, line 72-74): 

“As the population of SSA is the fastest growing and life expectancy is increasing most rapidly of all world regions...”

5. In table S1: Please explain what do you mean by the following terminologies: instable plaque, thrombus, reduced blood flow velocity.

Response: We agree with the reviewer that these terms were insufficiently defined. We amended the legend of Supplementary Table 1 as follows: 

“Supplementary Table S1. Carotid duplex sonography findings by stroke type:

IS = ischemic stroke; HS = hemorrhagic stroke

a Number and % if not indicated otherwise

b unstable atherosclerotic lesion at risk of rupture

c blood clot in carotid artery at risk of embolization

d post-stenotic reduced blood flow velocity in internal carotid artery”

6. Minor mistake: table 1 is described as figure 1.

Response: We agree with the reviewer that the journal’s requirement to insert a figure legend right after the paragraph in which the figure is cited can be confusing especially when a table and a figure legend are inserted together. We have changed the order of figure and table to improve readability of the manuscript. 

 

Reviewer #2: Very interesting well- written article that discussed the proportion of stroke types among Malagasy population. I have some comments that might help improve this nice article.

1. Line 164: (BMI) was more than 25 kg/m2, being an index it should not have units.

Response: We thank the reviewer for spotting this. We deleted the unit. 

2. Table 1: Length of hospital stay (days) (164, 104, 60) is a bit confusing → (days) should be (no of patients). Then (days) should be in the next rows, same applied for Length of hospitalization until death.

Response: We apologize for the inconsistency and have adjusted the rows in the Table 1 as suggested by the reviewer for improved clarity and readability.

3. Table 2: side of lesion was bilateral in 12.5% of cases however 70% of cases has Fazekas I -III which means that 70 % of patients have bilateral lesions. So, what do you mean here by bilateral lesions?

Response: We thank the reviewer for the opportunity to clarify this inaccuracy. The lesions we described in the first section of Table 2 refer to the stroke lesions. We found bilateral stroke lesions in 12.5% of cases. In addition to rating the stroke on CT, we included information on white matter lesions using the Fazekas Scale to provide additional information on chronic small vessel disease and cardiovascular risk [7]. We have amended Table 2 accordingly to clarify that we describe only stroke lesions in the first section of the table.

4. You did not mention the causes of death in your patients (chest infection, respiratory failure, pulmonary embolism, expanding hge/ inf etc). This is very important to know because your mortality rate is relatively high 22%.

Response: We fully agree with the reviewer and are grateful for the opportunity to provide more details about the causes of death in the study population. Unfortunately, data quality on the secondary causes was poor and did not allow to distinguish between different causes of death. We included the following paragraph in our results section (page 15, line 249-252):

“Most in-hospital deaths were directly attributed to the brain lesion (63.3%; 31/49, 95% CI: 49-77%). Secondary causes of death included chest infection, respiratory failure, and cardiac infarction.” 

5. What about the other important stroke risk factors like ischemic heart disease, oral contraceptive pills, migraine, atrial fibrillation, anticoagulants?

Response: We thank the reviewer for this comment. Unfortunately, details about these risk factors were documented very inconsistently in the available medical records. For most cases it remained unclear whether the information was missing, a risk factor had been ruled out or if screening had not been performed at all. Therefore, we decided to refrain from reporting on risk factors for which data was inaccurate or incomplete. 

6. You should mention in the limitation section that being a tertiary-level hospital this may cause selection bias as you will receive more of complicated and severe cases and this might explain the relatively higher mortality rate among your cohort.

Response: We fully agree with the reviewer that the selection of our study hospital might have contributed to a higher number of severe cases of stroke. We have included the following sentence in the limitations section of the manuscript (page 20 line 325-327):

“Second, the study hospital being a tertiary-level referral hospital might have introduced a selection bias towards more severe cases of stroke, which might explain the relatively high mortality rate. Third...”

 

Reviewer #3: 

This single center retrospective study using CT-confirmed strokes only, could miss a lot of ischemic strokes like minor strokes, posterior circulation strokes and others for which a CT scan might not have been less useful.

Even demographic and clinical data were not equally thorough and self-reported medical history may lead to an underestimation of negative findings.

Clinical data on stroke severity was limited and no standardized data on stroke severity was available.

All above mentioned might shed great doubt on the message of the study.

Response: We thank the reviewer for this comment. As also outlined in our responses to Reviewer 1, point 1 above we fully agree that it would have been preferable to include patients based on the clinical definition of stroke as well as to include more demographic and clinical data on stroke severity. However, overall accuracy and completeness of the available medical records for this retrospective analysis was poor. The points have been addressed in the limitations section of the manuscript with our initial submission (page 20, line 310-325): 

Our study has limitations. First, the results of our single-center study in an urban setting might not reflect the true community burden of stroke. However, access to healthcare including CT scans was free of charge at the study hospital for patients reducing selection bias otherwise introduced by a households' ability to pay. In addition, prompt imaging and comprehensive investigation would not have been feasible in a community-setting. Second, by including CT-confirmed strokes only, minor strokes for which a CT scan might not have been performed or posterior circulation strokes, for which CT imaging is known to be less sensitive, might be underrepresented. Third, while medical records including demographic and clinical data were available for all patients, they were not equally thorough and self-reported medical history may lead to an underestimation of negative findings. On the other hand, patients’ medical history was consistently assessed for relevant risk factors, like hypertension, in 96.9% of records. Fourth, clinical data on stroke severity was limited and no standardized data on stroke severity was available. Nevertheless, we used imaging characteristics to determine stroke severity. Last, our study was not designed to assess stroke prevalence or incidence, as the details of the source population from which the study hospital draws its patients from were unknown. 

Conducting research in a resource-restricted public health setting harbors additional challenges and limitations due to an overall lack of resources and poor data quality. In this manuscript, we draw from multiple data sources to best describe the proportion of stroke types in a hospital-based case-series in Madagascar’s capital Antananarivo, the only hospital in the country where patients can obtain a CT scan regardless of an individual’s financial situation. Despite a high burden of disease, scientific literature on stroke in Madagascar is scarce; there is a mere one-digit number of stroke studies from Madagascar. This lack of research draws an inconclusive picture, hampers efficient public health resource allocation in Madagascar, and adds to overall health inequity. There is an urgent need for more and better quality stroke data in SSA.

References: 

1. McLane HC, Berkowitz AL, Patenaude BN, McKenzie ED, Wolper E, Wahlster S, et al. Availability, accessibility, and affordability of neurodiagnostic tests in 37 countries. Neurology. 2015;85(18):1614-22.

2. Swindle RN, David. Barriers to Accessing Medical Care in Sub-Saharan Africa in Early Stages of COVID-19 Pandemic. Poverty and Equity Notes. Washington, DC. ©: World Bank; 2021.

3. Akinyemi RO, Ovbiagele B, Adeniji OA, Sarfo FS, Abd-Allah F, Adoukonou T, et al. Stroke in Africa: profile, progress, prospects and priorities. Nature Reviews Neurology. 2021;17(10):634-56.

4. World Bank. Fertility rate, total (births per woman): The World Bank Group; 2022 [Available from: https://data.worldbank.org/indicator/SP.DYN.TFRT.IN.

5. Wan He IA, Dzifa Adjaye-Gbewonyo. Africa Aging: 2020. U.S. Government Printing Office, Washington D.C. : U.S. Census Bureau; 2020.

6. Reutern G-Mv, Goertler M-W, Bornstein NM, Sette MD, Evans DH, Goertler M-W, et al. Grading Carotid Stenosis Using Ultrasonic Methods. Stroke. 2012;43(3):916-21.

7. Sharma R SS, Cascella M. White Matter Lesions. Treasure Island (FL): StatPearls Publishing; 2022. Available from: https://www.ncbi.nlm.nih.gov/books/NBK562167/.

---

## [Decision Letter · Decision Letter 1]

19 Sep 2022

PONE-D-22-13670R1Proportion of stroke types in Madagascar: a tertiary-level hospital-based case seriesPLOS ONE

Dear Dr. Emmrich,

Thank you for submitting your manuscript to PLOS ONE. After careful consideration, we feel that it has merit but does not fully meet PLOS ONE’s publication criteria as it currently stands. Therefore, we invite you to submit a revised version of the manuscript that addresses the points raised during the review process.

We look forward to receiving your revised manuscript.

Kind regards,

Ismail Ibrahim Ismail, MD

Academic Editor

PLOS ONE

Journal Requirements:

Reviewers' comments:

Reviewer's Responses to Questions

**Comments to the Author**

1. If the authors have adequately addressed your comments raised in a previous round of review and you feel that this manuscript is now acceptable for publication, you may indicate that here to bypass the “Comments to the Author” section, enter your conflict of interest statement in the “Confidential to Editor” section, and submit your "Accept" recommendation.

Reviewer #1: All comments have been addressed

Reviewer #2: All comments have been addressed

2. Is the manuscript technically sound, and do the data support the conclusions?

Reviewer #1: Yes

Reviewer #2: Yes

3. Has the statistical analysis been performed appropriately and rigorously? 

Reviewer #1: Yes

Reviewer #2: Yes

4. Have the authors made all data underlying the findings in their manuscript fully available?

Reviewer #1: Yes

Reviewer #2: Yes

5. Is the manuscript presented in an intelligible fashion and written in standard English?

Reviewer #1: Yes

Reviewer #2: Yes

6. Review Comments to the Author

Reviewer #1: Please remove the terminologies: "unstable plaque" and "thrombus" from table S1.

Actually, unstable plaque means plaque neovascularity or floating thrombus. I don't know if this is what the authors mean. I cannot understand what the authors mean by thrombus?

Reviewer #2: (No Response)

7. PLOS authors have the option to publish the peer review history of their article (what does this mean?). If published, this will include your full peer review and any attached files.

Reviewer #1: **Yes: **Ahmed Elhfnawy

Reviewer #2: No

---

## [Author Response · Author response to Decision Letter 1]

26 Sep 2022

Reviewer #1: 

Please remove the terminologies: "unstable plaque" and "thrombus" from table S1. Actually, unstable plaque means plaque neovascularity or floating thrombus. I don't know if this is what the authors mean. I cannot understand what the authors mean by thrombus?

Response: We thank the reviewer for pointing this out and apologize for the imprecise terminology in the previous version. We acknowledge that by non-contrast enhanced duplex ultrasound the diagnostic accuracy to assess the stability of a plaque and the sensitivity to detect intraluminal thrombi is limited. We report the findings as recorded by the Malagasy physician performing the ultrasound examination. To better describe the findings, we added details of the classification used to assess stability and clarified the term “thrombus”. 

The legend for “unstable plaque” now reads as:

“unstable atherosclerotic lesion at risk of rupture described as “echolucent” or “predominantly echolucent” corresponding to types 1 and 2 in the Gray-Weale classification [1].”

We omitted the term “thrombus” and specified to “intraluminal thrombus” which is also reflected in the legend now to read as: 

 “intraluminal carotid artery thrombus at risk of embolization”

---

## [Editor Report · Decision Letter 2]

2 Oct 2022

Proportion of stroke types in Madagascar: a tertiary-level hospital-based case series

PONE-D-22-13670R2

Dear Dr. Julius Emmrich

Kind regards,

Ismail Ibrahim Ismail, MD

Academic Editor

PLOS ONE
---

## [Editor Report · Acceptance letter]

5 Oct 2022

PONE-D-22-13670R2 

Proportion of stroke types in Madagascar: a tertiary-level hospital-based case series 

Dear Dr. Emmrich:

I'm pleased to inform you that your manuscript has been deemed suitable for publication in PLOS ONE. Congratulations! Your manuscript is now with our production department. 

Kind regards, 

on behalf of

Dr. Ismail Ibrahim Ismail 

Academic Editor

PLOS ONE